# Ponatinib and other CML Tyrosine Kinase Inhibitors in Thrombosis

**DOI:** 10.3390/ijms21186556

**Published:** 2020-09-08

**Authors:** Peng Zeng, Alvin Schmaier

**Affiliations:** 1Department of Pharmacology, Case Western Reserve University, Cleveland, OH 44106, USA; pxz147@case.edu; 2Departments of Medicine and Pathology, Case Western Reserve University and University Hospitals Cleveland Medical Center, Cleveland, OH 44106, USA

**Keywords:** ponatinib, tyrosine kinase inhibitors, Abl1 kinase, Bcr-Abl1, chronic myelogenous leukemia, thrombosis, platelet hyperactivity, pioglitazone

## Abstract

Abl1 kinase has important biological roles. The Bcr-Abl1 fusion protein creates undesired kinase activity and is pathogenic in 95% of chronic myeloid leukemia (CML) and 30% of acute lymphoblastic leukemia (ALL) patients. Targeted therapies to these diseases are tyrosine kinase inhibitors. The extent of a tyrosine kinase inhibitor’s targets determines the degree of biologic effects of the agent that may influence the well-being of the patient. This fact is especially true with tyrosine kinase inhibitor effects on the cardiovascular system. Thirty-one percent of ponatinib-treated patients, the tyrosine kinase inhibitor with the broadest inhibitory spectrum, have thrombosis associated with its use. Recent experimental investigations have indicated the mechanisms of ponatinib-associated thrombosis. Further, an antidote to ponatinib is in development by re-purposing an FDA-approved medication.

## 1. Introduction

Chronic myeloid leukemia (CML) causes ~4000 deaths annually in the US. As tyrosine kinase inhibitors (TKI) have been approved to treat CML, the overall survival from this disease has improved to 70%. With chronic treatment, the CML cancer stem cell constantly mutates, acquiring new resistance and necessitating new formulations of TKIs. As new compounds are introduced into the field with broader tyrosine kinase targets, the side-effects of these agents are increasing and have become problematic for some patients. This review covers the history of CML, the normal function of Abl1 kinase protein, Bcr-Abl1 pathogenesis, TKI development, and cardiovascular, especially thrombosis, side-effects of the TKIs. In this review, we will focus on prothrombotic effects of the third-generation TKI, ponatinib. We will discuss its prothrombotic mechanisms and a candidate non-anticoagulant antidote that could be combined with its use to prevent those thrombotic complications.

## 2. History of CML Biology and Research

CML is a hematologic stem cell disorder that presents with leukocytosis (i.e., elevated white-blood-cell count), with a large number and distribution of maturing myeloid progenitor cells (shift-to-the-left), increased basophils, and an enlarged spleen. It is molecularly identified by recognition of the Bcr-Abl1 (Breakpoint Cluster Region Protein-Abelson Tyrosine-Protein Kinase 1) fusion gene [1]. In 2016, there were 54,226 people living with CML in the USA [2]. There were about 4000 CML patient deaths in 2019 [3]. In 2020, 8450 new cases are expected [2]. CML was first reported and recognized in the 1840s [4,5]. Baikie and Tough described the so-called “Philadelphia (Ph) chromosome” in 1960 [6] and it was confirmed by Peter Nowell in 1962 [7] and Janet Rowley et al. in 1973 [8]. In 1986, Klein and Kleihauer published their characterization of the Bcr rearrangement and translocation of the cellular Abl1 oncogene in Philadelphia chromosome-positive leukemia [9]. In fact, the constitutively activated tyrosine kinase activity of the Bcr-Abl1 fusion protein results from Bcr chromosome 9 translocation to chromosome 22 t(9;22) (q34;q11), so Bcr becomes upstream of Abl1′s exon 2 [10]. This translocation activates myeloid cell growth and proliferation that signals through multiple oncogenic pathways, accounting for 95% of CML and 30% of acute lymphoblastic leukemia (ALL) patients [11,12]. 

Treatment of CML up to 1979 included cytotoxic agents starting with Fowler’s solution of arsenic, the use of radiation in the 20th century, and then chemotherapy with nitrogen mustard, hydroxycarbamide, and finally busulfan in the 1960s and 1970s. In 1979, the first identical twin bone marrow transplants were completed, and in the late 1980s, interferon-α was introduced. In the 1990s, bone marrow transplantation cured patients with CML. The first published use of a tyrosine kinase inhibitor (TKI) was in 1998 [13]. TKIs, for example, imatinib, acts as a competitive inhibitor maintaining closed conformation and precluding ATP binding of Abl1 kinase, thus shutting down the proliferation stimulus. This class of agents has made a lethal disease into a chronic medical illness. 

## 3. Normal Function of Abl1 Tyrosine Kinases

The human Abl1 gene was first discovered as a Bcr-Abl1 fusion gene in a CML patient by David Baltimore in 1986 [14]. His laboratory was one of the two groups that identified its kinase activity [15,16], and characterized a link between the Ableson virus and cellular Abl kinase [17]. Ableson discovered a murine lympho-sarcoma virus in 1970. The gene Abl1 was named after him [18]. Abl2 or Arg was identified as a paralog by sequence homology 20 years later [19].

The functions of Abl1 kinase have been depicted in several reviews [20,21]. The structure of Abl1 includes a cap sequence, SH3, SH2, kinase domain, DNA binding domain, G-actin binding domain, and F-actin binding domain [20]. Myristoyl displacement of the cap sequence by the kinase domain initiates the releasing of Abl1 from autoinhibition. SH3 and SH2 domains stabilize the auto-inhibitory inactive state of Abl1 kinase. Tyrosine phosphorylation of Y^245^, localized in the linker of SH2 and the kinase domain, and Y^412^, localized in the activation loop of the kinase domain, are associated with increased kinase activity. It is unclear if Abl1 kinase has tyrosine phosphorylation gating of its kinase activity like Src family kinase [21]. Abl1 gets transported into the nucleus to impact transcription through its DNA binding domain. Its G or F actin binding domains are the direct structural basis for its role in cell motility and adhesion. 

There are 116 Abl1-interacting proteins, 76 of them are substrates of and bind to Abl. Those proteins are functionally diverse, including adaptors, other kinases, cytoskeletal proteins, transcription factors, chromatin modifiers, etc. [20]. In addition to its kinase function, Abl1 also functions as an adaptor (i.e., for cytoskeleton proteins) and an RNA processing factor. Genetically manipulated mouse lines have demonstrated tissue-specific functions of Abl1 kinase.

Abl1 is an essential gene for viable mammalian life. Abl1 knockout mice have perinatal lethality with cardiac hyperplasia. They are rescued by the cardiomyocyte-specific installation of Abl1 [22,23]. Abl1 and 2 double knockout mice are embryonically lethal [24]. In myeloid cells, constitutively active Abl1 kinase as seen in CML leads to cell growth and proliferation. A murine tie2-cre-driven endothelial cell knockout of Abl1 in an Abl2^-/-^ background dies at late embryonic or perinatal stages with downregulated endothelial angiopoietin/tie2 signaling, increased endothelial apoptosis, and an abrupt end to endothelial survival [25]. Thus, in the endothelium, Abl1 kinase is essential for survival and vascular integrity. In T-cells, Abl1 is important for maintenance of cellular junctions and HEF1 and GTPase in cell migration. Its influence on cell migration derives from its activation of growth factors and chemokines, leading to cadherin and integrin movement to regulate cell homeostasis, cytoskeletal remodeling, and adhesion [26,27]. A conditional knockout of Abl1 in T cells (Lck-Cre) results in impaired cell-cell interaction and motility [26,28]. Abl1 also regulates tight junctions in endothelial cells [27]. In smooth muscle, conditional knockout mice with a Sm22-cre have hyper-responsive airways in asthma models, suggesting that the gene balances smooth muscle tone [29]. Likewise, in skeletal muscle, conditional KO mice produced with a MyoD-cre have impaired DNA-damage repair [30]. 

The abundance of the Abl1 kinase is regulated by ubiquitination [31,32]. In vitro de-ubiquitinase inhibitor experiments show that the Bcr-Abl1 fusion protein is rapidly modified by Lysine63-linked ubiquitin polymers [33]. Dephosphorylation of Abl1 Y^245^ is performed by PTPN12 and PTPN18 [34,35]. PTPN12 knockouts in a *Drosophila* screening model were observed to be phenotypically resistant to TKIs used to treat CML [36], whereas other cellular phosphatases like PTPN1 (PTP1B), PTPN6 (SHP-1), and PTPN11 (SHP-2) are substrates of Abl1 kinase. They have been reported to promote Bcr-Abl1-induced hematologic neoplasia (CML and B cell acute lymphoblastic leukemia (ALL)) by different groups [37,38,39,40,41].

## 4. Pathogenic Function of Bcr-Abl1 Kinase Fusion Proteins

Bcr is a serine/threonine kinase with several interaction domains for proteins such as actin, lipids, and GTP [42,43,44]. In Bcr-Abl1-positive CML and ALL patients [11,12], Abl1 is a constitutively activated tyrosine kinase. The upstream location of Bcr to Abl1 kinase is the genesis of activity [10]. Moreover, different segmental translocations lead to distinct forms of Bcr-Abl1 fusion proteins expression, which are p185, p210, and p230. P210 is most common, causing CML, while the other two are associated with neutrophilic leukemia (p230) and ALL (p185), respectively. 

It is unclear if Bcr-Abl1 is a somatic (acquired) or germline (inherited) mutation. First, experimental hybridization of chromosome 9 Bcr with chromosome 22 Abl1 has been done in mice and patient somatic cells [45]. Second, the incidence of the Bcr-Abl1 fusion gene in healthy people is age-related, which is 2% (*n* = 44) in 0–13 years old and 30% (*n* = 73) in 20–80 years old [46]. 

Additionally, the Bcr-Abl1 fusion gene is not sufficient for CML development. Some pre-leukemia somatic mutations, such as epigenetic genes, are required for the transformation [47,48,49,50]. There has not been extensive screening of healthy individuals to determine who carries the Bcr-Abl1 translocation and, if treating them, makes a difference in outcomes [44]. Alternatively, TKI-targeting Bcr-Abl1 in patients with CML or ALL have brought responsive patients a close-to-normal life span [1]. The Bcr-Abl1 fusion protein ultimately activates myeloid cell growth and proliferation that signals through multiple oncogenic pathways [11,51,52].

## 5. Mutations in Bcr-Abl1 Fusion Protein Have Led to the Development of Several TKIs

As the first small molecule Bcr-Abl1 targeting TKI imatinib became available in 2002, the five-year survival of the CML patients increased from 20–30% (1989–2001) to 50–90% (2001–2013) [53,54,55,56,57,58,59,60]. TKIs used in CML management, with the exception of asciminib (binds a myristoyl site of the BCR-ABL1 protein, “locking” BCR-ABL1 into an inactive conformation via a mechanism other than binding to the kinase ATP-binding site), target the ATP binding pocket in the Abl1 kinase. The ATP binding pocket is well-conserved among protein kinases. The variabilities in this domain have an important role in determining the affinity between it and a specific TKI [61,62]. Based on in vitro cell proliferation assays, a spectrum of targets for each approved human use TKI is known [63]. In the present report, we extracted these data to prepare a table of targets in vascular biology and platelet activation for each of the FDA-approved TKIs used to manage CML (Table 1).

All the approved TKIs inhibit Abl1 kinase. The first TKI imatinib has the smallest spectrum of tyrosine kinase targets. Akt, Erk family members, FAK, and CDK4 pathways are spared from these TKIs, allowing their signaling cascades in tissues. Ponatinib’s universal inhibition on FGFR, VEGFR, and PDGFR family members suggests that it may have a large effect on the cardiovascular system as these growth factors influence cell growth, proliferation, angiogenesis, reperfusion, repair, and hypertrophy and fibrosis. Ponatinib also uniquely inhibits FGFR4 and Tie2. Regarding platelets, the CSK and CHK family phosphorylate C terminal tyrosines to inhibit Src family kinases’ activity. Their inhibition may result in activation of Src family kinases (Src, Yes, Fyn, Fgr, Lck, Hck, Blk, Frk, and Lyn) [64,65,66]. Among them, Lyn and Fyn are an important in signaling and regulation of platelets function [67,68]. However, as Src family kinases have two critical tyrosine sites that have opposite roles for their kinase activity, the ability of TKI targeting a Src family kinase to activate or inhibit the enzyme depends on the concentration and specificity of the inhibitor [64,69]. Dose dependency of a TKI’s effect may allow an agent like ponatinib to activate platelets by manipulating Lyn in the GPVI activation pathway [64]. It is also worth noting that, in the vessel wall, Tie2 signaling is also critical for endothelial survival [70]. Ponatinib’s inhibition of Tie2 may lead to vessel wall injury [25]. In one word, the sum of ponatinib’s inhibition spectrum may result in a prothrombotic phenotype in vivo. Ponatinib’s association with cardiovascular events, especially clinical and experimental thrombosis, will be discussed below in Section 5 and Section 6. 

Another aspect of TKI use is that with treatment, patients develop resistance to the agent. This resistance is due to the development of new mutations in the Bcr−Abl1 fusion gene. Table 2 is a list of the mutations in Bcr−Abl1 that have been observed with continued TKI treatment [51]. A large number of mutations arose after use of the first−generation TKI imatinib that produces resistance to the agent. A common feature of those mutations is that they are near or in the imatinib binding site. There are a large number of mutations at the ATP binding pocket at the SH2 contact, C−lobe, the activation loop, and a few at gate keeper residues (where ATP accesses the kinase domain and mediates the conformation change from close to open). Among them, the most resistant is the *T315I* mutation that makes the Bcr−Abl1 kinase’s ATP binding pocket inaccessible to imatinib, nilotinib, bosutinib, and dasatinib. The *T315I* mutation is estimated to be as high as 19% in the general population [71,72,73,74,75]. 

Multiple strategies to get around the *T315I* mutation have been unsuccessful until the agent ponatinib was developed [76]. A TKI candidate ONO1230 targets Crk, the first substrate of Bcr−Abl1. It exhibited a 10−fold increased potency compared to imatinib including *T315I*, but its development was stopped in preclinical investigations [77]. Second−generation TKIs (dasatinib and nilotinib) exhibited overall significantly higher and faster rates of complete cytogenetic response and major molecular responses than imatinib in the clinic trials [78,79,80]. However, the *T315I* mutation still blocks these agents [81,82]. Two newer agents, however, ponatinib and asciminib, are able to target the *T315I* mutation (Table 2) [51,76,83].

## 6. The Use of TKIs in CML and Their Association with Cardiovascular Disease

The observation that ponatinib was not inhibited by the *T315I* mutation made it a prime agent for management of patients with this polymorphism and those patients who became resistant to other TKIs. There have been numerous clinical trials evaluating the efficacy of ponatinib [84,85,86,87,88,89,90]. One initial trial of 29 patients reported no thrombotic events recorded during a median follow−up of 12 months [87]. Another study of 37 patients recorded one patient having a vascular adverse event for a median follow−up of 14.8 months [90]. In a third investigation of 62 patients, 11 of them (18%) had thrombotic events after a median time of 5.8 months with ponatinib use and a median follow up of 26.5 months [89]. However, the five−year follow−up of the pivotal Phase−II Ponatinib Ph+ ALL and CML Evaluation (PACE) trial showed a cumulative 31% of arterial and venous occlusive events out of 449 ponatinib−treated CML patients. Importantly, the exposure−adjusted incidence of new arterial occlusive events decreased over time (15.8 and 4.9 per 100 patient−years in year 1 and 5, respectively) [88]. This pattern of ponatinib inducing thrombosis is confirmed in a thrombosis−focused CML study [85].

The Evaluation of Ponatinib versus Imatinib in Chronic Myeloid Leukemia (EPIC) study was a randomized, open−label, phase 3 trial designed to assess the efficacy and safety of ponatinib, compared to imatinib, in newly diagnosed CML patients from 106 centers in 21 countries. Eleven of 155 ponatinib patients (7%) had occlusive arterial events versus 3 (2%) of 152 patients given imatinib (*p* = 0.052) [86]. However, arterial occlusive events in 6 of the 11 ponatinib−treated patients and 1 of the 152 imatinib−treated patients (*p* = 0.010) were considered serious. Consequently, this Phase−III EPIC study was terminated prematurely due to concern for safety [86,91]. 

The cardiovascular events associated with TKI use was not initially appreciated by the CML−treating physicians. The ponatinib findings prompted investigators to do similar analysis on other populations of treated CML patients. Several reviews have summarized the vascular toxicity of TKI therapy [92,93]. Recently, an FDA analysis of the adverse event reporting system database showed that among 64,232 reported cardiovascular events of cancer patients, 2678 of them are CML patients managed with TKIs. The cardiovascular effects of the TKI used to treat CML were analyzed and are summarized in Table 3 [94]. Determined by the odds ratio, nilotinib treatment is associated with cardiac arrhythmias. Cardiac failure is associated with nilotinib, dasatinib, bosutinib, and ponatinib. Embolic and thrombotic events occur in both nilotinib and ponatinib patients. Hypertension is associated with ponatinib management. Ischemic heart disease is seen with nilotinib, bosutinib, and ponatinib treatment. Significant pulmonary hypertension is associated with imatinib and dasatinib. Lastly, QT prolongation is seen in nilotinib and dasatinib patients (Table 3). 

Although some data show that nilotinib use is associated with a prothrombotic phenotype in vivo and in vitro [95,96], the clinical trial data do not support that. A three−year follow up of the randomized Evaluating Nilotinib Efficacy and Safety in Clinical Trials Newly Diagnosed Patients (ENESTnd) data reported 5 of 279 (300 mg twice daily) and 3 of 277 (400 mg twice daily) patients had peripheral arterial occlusive events [79], which is not specifically reported in the five−year update [80]. The update reported no deep vein thrombosis, and 1 patient of 279 (300 mg twice daily) had retina vein occlusion [80]. A recent Phase II Nilotinib With Newly Diagnosed Chronic Phase CML trial (NCT00129740) that was performed by a review of adverse events did not show any increased incidence of thrombotic adverse events for 148 patients with a dose of 400 mg twice daily [97]. In a further assessment of the potential problems of TKI use in CML and cardiovascular events (CV), a review of six clinical trials that includes 531 patients was performed looking for CV adverse events (AE) in patients who received frontline agents [98]. Overall, 237 patients out of 531 (45%) developed CV−AEs. Hypertension was the most common AE in 175/531 (33%) with 17% having a severe grade 3 out of 4 blood pressure elevation. The CV−AE incidence ratios (IRs) with a 95% confidence interval are 8.6 (7.6–9.8) per 100−person years [98]. Among the TKIs, ponatinib shows the highest IR at 40.7 (27.9–59.4). Concerning CV thrombosis (myocardial infarction, stroke, peripheral vascular occlusion, and carotid artery occlusion), ponatinib had an incidence of 7/43 (16.3%) versus imatinib 13/274 (4.7%), nilotinib 10/108 (9.3%), and dasatinib 17/106 (16%). These data indicate that adverse CV events are not the exclusive domain of ponatinib.

## 7. Mechanism(s) of Ponatinib−Induced Thrombosis

The mechanism of how ponatinib induces thrombosis is not completely understood [96,99,100]. Loren and colleagues investigated the possible prothrombotic effect of ponatinib in 2015. The in vitro addition of 0.1 to 1 µM ponatinib to washed platelets inhibited GPVI activation pathways when compared to untreated samples [99]. In addition, ponatinib−treated platelets spread less on fibrinogen or collagen surfaces similar to platelets treated with the Src family kinases inhibitor PP2, but in contrast to 1 µM nilotinib− or imatinib−treated samples. Second, ponatinib−treated platelets attached to fibrinogen or collagen have reduced phosphorylated GPVI pathway kinases such as LynY^507^, SrcY^416^, LatY^191^, or Btk Y^223^ similar to PP2−treated platelets, but not nilotinib− or imatinib−treated samples. Third, platelet aggregation in response to stimulation with collagen−related peptide (CRP) (3 μg/mL) was dramatically decreased by ponatinib at concentrations as low as 0.1 µM, a concentration that is close to maximum plasma concentration observed in patients [99,101]. These platelets have a 4.3 ± 20.5 percent aggregation and a lag time of 42.0 ± 5.1 s, compared to vehicle−treated platelets (1% DMSO) that have 100 percent aggregation and 14.3 ± 3.1 s lag time (*p* < 0.05). In these investigations, ponatinib treatment significantly decreased the percent aggregation and increased the lag time for platelet shape change and aggregation in response to CRP stimulation. Lastly, ponatinib treatment also decreases platelet P−selectin and phosphatidylserine exposure, as detected by flow cytometry, in response to collagen−related peptide (CRP = 10 μg/mL, *n* = 3, *p* < 0.05). These combined studies indicate that integrin α_2b_β_3_ and GPVI pathways are inhibited in ponatinib−treated (0.1 to 1.0 μM) platelets.

Hamadi and colleagues have also examined ponatinib use ex vivo and in vivo [96]. Using 0.1 and 1 µM ponatinib in an ex vivo flow model on a collagen surface, ponatinib treatment promoted thrombus growth. Additionally, they showed that with administration of 3 mg/kg ponatinib orally to 8 week old C57BL/6 mice, the ponatinib−treated mice were prothrombotic in ferric chloride assay with both a significantly shortened time for occlusion and larger thrombus volume. They also detected increased inflammatory cytokines TNFα and IL6 at 4 h after ponatinib treatment. Moreover, they observed that administration of the calcium channel blocker diltiazem 24 h before ponatinib is given reduced the prothrombotic effect of this TKI. 

Having the result of the PACE trial, we independently examined the prothrombotic effect of ponatinib [100]. When these investigations began, we were aware of the Loren et al. study and we chose to create an in vivo model that mimics a steady state of ponatinib use with plasma concentration of the agent in wild−type C57 mice similar to that seen in man. We treated 18–20 week old mice (median age of CML patients is 60 years old) with 3 mg/kg orally twice a day of ponatinib for 14 days. Drug levels for ponatinib at 3 mg/kg/po bid were obtained by LC/MS/MS assay on day four, 2 and 24 h after starting treatment. They were 176 ± 47 and 11.4 ± 3.4 ng/mL, respectively, which translates to maximal final in vivo concentrations of 33 to 2 nM, respectively [100]. These concentrations are similar to those achieved in man by oral administration [102]. Initial investigations show that mice dosed that way have significantly shorter times to carotid artery occlusion following thrombosis induction with Rose−Bengal compared to vehicle (citrate buffer)−treated mice (10.4 ± 2.9 min versus 32.3 ± 4.8 min, *p* < 0.0001). We did not see the same shortening of carotid artery thrombosis times in 8–10 week old C57BL/6 mice with similar ponatinib treatment. Once establishing that ponatinib−treated mice are having shortened provoked occlusion times, we embarked upon an investigation to determine if the prothrombotic state was due to changes in murine blood coagulation proteins, the vessel wall, or platelets, the three critical components that determine the balance between hemostasis and thrombosis [103]. 

Ponatinib−treated animals have normal prothrombin (PT) and activated partial thromboplastin (aPTT) times and are not different from vehicle−treated mouse plasmas. In addition, after ponatinib treatment, contact activation and tissue factor−mediated thrombin generation times (TGT) show no statistical difference between ponatinib−treated and vehicle−treated mouse plasma. These data indicate that blood coagulation proteins were not aberrant in ponatinib−treated mice. Additionally, ponatinib−treated mice have a normal complete blood count including platelet count and normal white blood cell differential cell count [100]. 

As ponatinib uniquely inhibits a number of signaling systems that influence the vessel wall (FGFR, VEGFR, PDGFR, and Tie2) (Table 1) and inhibition of Abl1 kinase negative effects in endothelium [25,27], investigations examined if steady−state levels of ponatinib influence vessel wall biology [104,105,106]. Investigations determined if the aorta from ponatinib−treated mice show evidence of apoptosis and increased reactive oxygen species. Vessel wall apoptosis is associated with thrombosis [107]. In murine tissue investigations, we found significantly increased expression of caspase3 in vessel adventitia including adipose tissue [100]. We also identified increased reactive oxygen species (ROS) using the antibody to nitrotyrosine in the ponatinib−treated vessel wall [100]. Finding vessel apoptosis of the adventitia is associated with increased nicotinamide adenine dinucleotide phosphate oxidase−derived ROS [108,109].

Additional investigations examined if ponatinib influences platelet function to determine if they also contribute to the observed arterial thrombotic events seen in patients. Tail bleeding times show that ponatinib−treated mice are 55 ± 12 sec (mean ± SEM), a value shorter than vs. 102 ± 9.3 sec for untreated mice, *p* < 0.033, *n* > 20 in each group [100]. We next examined platelet GPVI activation after collagen−related peptide (CRP) and protease−activated receptors after α−thrombin stimulation. In vivo ponatinib treatment primed platelets such that their CRP or α−thrombin−induced expression of JON/A (the epitope of the activated heterodimeric complex of α_2b_β_3_ integrin complex on murine platelets) (*p* < 0.01) and P−selectin (CD62) (*p* < 0.01) was significantly higher than that of untreated platelets when examined on flow cytometry [100]. These studies indicate that in vivo administration of ponatinib alters platelets such that they become activated at lower concentrations of CRP and α−thrombin [100]. The difference between these experiments where platelets are treated in vivo with ponatinib versus the in vitro investigations of Loren et al. [99] is that the final concentration of the TKI was between 2–33 nM in the in vivo experiments versus 100–1000 nM in the in vitro studies [99,100]. As Lyn is constitutively active, the lower dose in the in vivo studies may have produced the open conformation of p−Lyn by blocking only the higher inhibitory site at p−LynY^507^ [101]. These combined studies indicate that ponatinib treatment alters both vessel wall and platelet function, making the latter hyperactive. 

Next, we sought an antidote to ponatinib’s effect on the vessel wall and platelets. Preliminary studies indicated that in ponatinib−treated mice, aortic Sirt1, KLF4, thrombomodulin, and endothelial cell nitric oxide synthetase were reduced. We postulated that a PPAR−gamma agonist, i.e., pioglitazone, that stimulates Sirt1 might correct these defects. When ponatinib−treated mice are also given pioglitazone (10 mg/kg/day, orally), the time to thrombosis significantly lengthens and corrects to normal (41 ± 3.7 min, *p* < 0.025) compared to ponatinib treatment alone (*p* = 0.0009) [100]. Additionally, combined ponatinib and pioglitazone treatment corrects the increased ROS and apoptosis in murine aorta as indicated by no nitrotyrosine and caspase 3 expression, respectively [100]. Further, pioglitazone treatment combined with ponatinib normalizes (i.e., increases) the threshold concentration of CRP to induce platelet JONA and P−selectin (CD62) expression on platelets [100]. These experiments indicate that in our murine model, pioglitazone is an antidote to ponatinib. In several randomized clinical trials, pioglitazone has been recognized to be protective of cardiovascular events [110,111,112]. Additionally, as pioglitazone is a Stat5 inhibitor, it has been shown that it also increases molecular remission in CML [113,114,115]. We observed Stat5 inhibition in our experiments [100]. 

## 8. Summary and Perspective

In the last 20 years, the discovery and use of TKIs to treat CML have been a major success. With continued use of TKIs, it has been recognized that some are associated with negative cardiovascular events. This effect is clearest for ponatinib. However, understanding the mechanism(s) of ponatinib−induced thrombosis will guide us to re−purpose or develop novel therapies to manage the TKI toxicities so that these effective anti−cancer agents continue to be used. 

## Figures and Tables

**Table 1 ijms-21-06556-t001:** Tyrosine kinase inhibitors (TKI) specificity and extent of inhibition.

	Imatinib	Nilotinib	Bosutinib	Dasatinib	Ponatinib
ABL1	83	98	100	105	101
ABL2	68	95	99	102	100
AKT1	3	11	−5	3	9
AKT2	4	12	−3	6	7
AKT3	4	2	1	5	16
BLK	23	29	84	103	97
BTK	−1	45	97	102	95
CDK4/CycD3	3	−4	−7	3	24
CHK1	10	−15	83	11	3
CHK2	6	−12	87	4	96
CSK	−2	78	84	104	102
EEF2K	−2	6	−7	3	−4
EGFR	4	17	100	102	97
EPHA1	9	61	3	101	97
EPHA4	5	91	86	99	101
EPHB1	7	72	98	100	100
Erk1	−5	3	1	−5	−2
Erk2	−8	1	−4	−7	−7
Erk5	17	−4	3	5	3
FAK	29	3	22	17	−23
FGFR1	−1	−29	79	47	101
FGFR2	3	−67	95	73	100
FGFR3	1	−11	83	34	101
FGFR4	8	−7	3	9	98
FGR	28	55	92	103	101
VEGFR1	5	32	97	39	101
VEGFR2	7	22	101	22	94
VEGFR3	3	17	92	31	101
FLT3	68	60	77	17	99
FRK	9	70	94	100	100
FYN	30	59	95	100	101
HCK	13	73	89	100	98
JAK1	−1	13	−2	9	99
JAK2	0	19	64	68	92
KIT	97	96	23	100	101
LCK	80	90	101	102	103
LYN	76	85	93	100	100
PDGFRα	98	103	77	100	103
PDGFRβ	91	93	95	99	102
SRC	5	23	96	101	102
SYK	16	54	100	69	10
TIE2	0	41	22	16	101
YES	22	44	97	102	101
ZAP70	10	3	76	12	5

These data were extracted and put into the present format from Supplemental Table S6 from Reference [63]. See text for explanation. The “red” background means higher inhibition; the “green” background means lower inhibition. A minus number means increased kinase activity.

**Table 2 ijms-21-06556-t002:** TKI−resistant mutations observed in patients with chronic myeloid leukemia (CML).

TKIs	Imatinib	Nilotinib	Bosutinib	Dasatinib	Ponatinib	Asciminib
Binding conformation	closed	closed	Both	Open	Closed	Myristoyl pocket
Resistance	T315 Y253 E255 M244 L248 G250 Q252 F317 M351 M355 F359 H396	T315 L248 Y253 E255 F359	T315 V299 L248 G250 E255 F317	T315 V299 F317	E255	A337 W464 P465 V468 I502

These data were extracted from Figure 3 of Reference [11].

**Table 3 ijms-21-06556-t003:** CML TKIs and their risk for adverse cardiovascular events.

TKIs/Syndromes	Imatinib	Nilotinib	Bosutinib	Dasatinib	Ponatinib	Asciminib
Cardiac arrhythmias	0.3(0.1–1.4)	2.7(2.1–3.5)	1.6(0.2–11.7)	1(0.6–1.6)	1(0.5–2.2)	NA
Cardiac failure	1.1(0.8–1.6)	1.5(1.3–1.7)	3.5 (1.9–6.6)	4.1(3.7–4.6)	1.8(1.4–2.4)	NA
Cardiomyopathy	1.2(0.7–2.0)	0.4(0.2–0.6)	NA	0.4(0.3–0.7)	0.6(0.3–1.2)	NA
Embolic, thrombotic	0.4(0.3–0.5)	1.3(1.1–1.4)	1(0.5–1.9)	0.5(0.4–0.6)	1.4(1.2–1.6)	NA
Hypertension	0.2(0.1–0.5)	0.9(0.8–1.1)	1.2(0.4–3.7)	0.8(0.6–1.0)	3.5(2.9–4.3)	NA
Ischemic heart	0.6(0.4–0.9)	6.7(6.2–7.2)	2.5(1.3–4.8)	1.0(0.8–1.2)	2.9(2.4–3.5)	NA
Pulmonary hypertension	3.9(2.4–6.4)	1.1(0.6–1.7)	NA	8.5(6.8–10.6)	1.3(0.6–3.9)	NA
QT prolongation	0.8(0.6–2.5)	12.2(10.3–14.6)	NA	2.5(1.6–3.7)	0.9(0.3–2.4)	NA

The table presents an adjusted reported odds ratio to a specific adverse cardiovascular event. A “green background” means there is a protective effect of the drug. A “red” background means that the drug use was associated with an adverse cardiovascular effect. These data were extracted and tabulated presently from information in Supplementary Tables S1–S8 presented in Reference [94].

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
