# Peer review of "Ponatinib and other CML Tyrosine Kinase Inhibitors in Thrombosis"

_ijms, 2020, doi:10.3390/ijms21186556_

Round 1

Reviewer 1 Report

The authors presented a detailed review of the application of TKI in patients with CML. The manuscript is well presented and the English language well reported. The limit is represented by the lack of statistical details and selection of the articles which would have been well illustrated by the realization of a systematic review. Moreover, further important clarifications are however necessary on many points of the discussion.

Line 121. Please better explain the role of TK in vessels homeostasis, including migration.

Line 179. the main side effects on the cardiovascular system and in particular on the natural history of aortic aneurysms should be described.

Line 191. the authors should illustrate whether in the literature there is evidence of thrombotic events in vivo and what are the molecular and macroscopic mechanisms underlying them.

Line 231. please explain if possible the side of thrombotic events and related surgical or pharmacological approach to solve the problem. It is also useful to report the long-term prognosis data of affected patients and the primary patency.

Line 255. What were the common cardiovascular comorbidities? if possible indicate the use of drugs with systemic anti-inflammatory action.

Line 268. the authors should explain how the general thrombotic risk was assessed in the studies cited. Has any particular score been used?

Line 301. please explain more fully the role of inflammatory cytokines in triggering the thrombotic process. Furthermore, the authors should explain the possible mechanism of negative regulation of the homeostasis of the inflammatory process by endogenous inhibitors, if present.

Line 339. please explain if there are in vivo works in the literature and in particular on humans

Line 371. Please illustrate whether the administration of pioglitazone resulted in side effects in the treated subjects.

Author Response

Thank you for your time spent on reviewing our article.  We appreciate your constructive comments that have helped to make the manuscript clearer.  Below are our edits to each of your comments,  

Q: Line 121. Please better explain the role of TK in vessels homeostasis, including migration.

A: We appreciate the comment. This section has been revised.  See lines 119-121 of the revised manuscript.

Q: Line 179. the main side effects on the cardiovascular system and in particular on the natural history of aortic aneurysms should be described.

A: These growth factors influence cell growth, proliferation, angiogenesis, reperfusion, repair, and hypertrophy and fibrosis.  This information has been added to the text on revision lines 178-180.  On literature review (Pubmed) we were not able to find a specific reference in the experimental or clinical literature indicating that ponatinib has a direct influence on aortic aneurysm development.  Certainly, its influence on hypertension may indirectly lead to aortic aneurysm development.  However, we believe that a detailed discussion here on aortic aneurysm development presently is tangential to the present topic.

Q: Line 191. the authors should illustrate whether in the literature there is evidence of thrombotic events in vivo and what are the molecular and macroscopic mechanisms underlying them.

A: Thank you for identifying the main point of this manuscript.  In Section 5 below we describe the clinical data showing that ponatinib use is associated with a 31% accumulative incidence of thrombosis.  See references 85,88,89.  Further in Section 6, our experimental data (reference 100) and those of others (95,96) show various vessel wall and platelet activation states induced by ponatinib and nilotinib in murine models.  We now point the reader to those Sections on lines 193-195 in the revision.

Q: Line 231. please explain if possible the side of thrombotic events and related surgical or pharmacological approach to solve the problem. It is also useful to report the long-term prognosis data of affected patients and the primary patency.

A: This question is difficult to answer because there are no prospective trials examining treatments to prevent cardiovascular events associated with ponatinib or any other TKI use.  This content area is newly recognized.  Because of the cardiovascular toxicity of ponatinib, oncologists only use ponatinib when a patient with CML and ALL presents with the T315I mutation in BCR-ABL or when the patient fails all other TKIs treatment for their hematologic malignancy.  Long-term prognosis is based on management of the cancer, not drug-induced cardiovascular events.  There are relatively few patients on this medication.  The medical oncology community has managed these complications by lowering the ponatinib dose or with aspirin.  Occasionally when a venous thrombosis is demonstrated, an anti-factor Xa anticoagulant (DOAC, enoxaparin) is given.  Our experimental data (Reference 100) indicate that the PPARgamma agonist, pioglitazone, effectively removes the thrombosis risk by eliminating the vascular inflammation and hyperactive platelets induced by ponatinib in murine models of thrombosis and platelet investigations.  Note, in our murine arterial thrombosis model, aspirin is ineffective (unpublished).

Q: Line 255. What were the common cardiovascular comorbidities? if possible indicate the use of drugs with systemic anti-inflammatory action.

A: Thank you for asking this question.  Table 3 indicates exactly the cardiovascular events associated with each of the currently used TKIs.  Reference to Table 3 now is made in this sentence in the revised manuscript line 255.  It is currently not possible to indicate what anti-inflammatory drugs ameliorate the effects of TKIs.  For the review purpose only, we have observed on RNAseq experiments on murine aorta that combined treatment of pioglitazone with ponatinib versus ponatinib alone reduces to normal ~100 gene sets that had been upregulated by ponatinib.  These gene sets include inflammatory markers like cytokines, interleukins, etc. (unpublished).  We are preparing this research for future publication.

Q: Line 268. the authors should explain how the general thrombotic risk was assessed in the studies cited. Has any particular score been used?

A: There was no prospective scoring system in the investigation in references 97 and 98.  CV events were recognized as adverse events (AEs) or serious adverse events (SAEs) by the clinicians in the trial.  The analysis was retrospective. 

Q: Line 301. please explain more fully the role of inflammatory cytokines in triggering the thrombotic process. Furthermore, the authors should explain the possible mechanism of negative regulation of the homeostasis of the inflammatory process by endogenous inhibitors, if present.

A: This area needs further investigation.  The authors of reference 96 observed TNFa and IL6 elevation 4 h after ponatinib treatment.  In unpublished investigations that we are working on, aortic RNAseq data reveal that interleukins and cytokines are the third and fifth gene sets that are most up-regulated after ponatinib therapy.  We are currently actively involved in these investigations.  It appears that the use of ponatinib itself initiates an inflammatory process that is capable of elevating IL6, TNFa and many other interleukins and cytokines (unpublished data in progress).

Q: Line 339. please explain if there are in vivo works in the literature and in particular on humans.

A: These findings of increased caspase3 and ROS expression after ponatinib treatment has not been identified to our knowledge in human tissue.  Reference 106 indicates the vessel wall apoptosis from any cause is associated with thrombosis.  A recent preprint on zebrafish treated with ponatinib indicates increased caspases and interleukins (See Zhu X-Y et al. Ponatinib-induced ischemic stroke in larval zebrafish for drug screening. Eur J of Pharmacology, In Press doi: https://doi.org/10.1016/j.ejphar.2020.173292.

Q: Line 371. Please illustrate whether the administration of pioglitazone resulted in side effects in the treated subjects.

A: We did not use pioglitazone in any human subjects.  Pioglitazone use in the IRIS trial (Young LH et al. Circulation 138:1210, 2018) is associated with weight gain, edema, and shortness of breath when slowly titrated in patients from 15 mg po daily to 45 mg po daily over 2 months.  In the 2 weeks of murine treatment, we did not observe any pioglitazone-related side effects in mice, but the experimental investigation was not designed to examine for side effects.  There are no published human studies with pioglitazone for ponatinib-induced thrombosis.  However, pioglitazone had been given to CML patients to induce molecular remission (Rousselot P et al. Cancer 123:1791, 2017).  Pioglitazone is also a Stat5 inhibitor which itself induces molecular remission.  There was no change in the CBCD between treated and untreated.  No change in HbA1C and total cholesterol, and no instances of hypoglycemia.  Patients weights were overall stable although 12 patients out of 24 (50%) did gain weight.  One patient out of 24 had significant edema.  All subjects were screening for bladder cancer at 6 and 12 months and none were found. 

Reviewer 2 Report

The authors present an interesting review on the area of CML/ALL and chemotherapy related thrombosis. The review is more comprehensive about the biology surrounding CML and ALL rather than discussing the chemotherapy induced thrombosis however. Tables and diagrams complement the review well and are well presented.

More discussion is warranted on the side of thrombsis. Cancers themselves are extremely pro-thrombotic, with cancer associated thrombosis the 2nd leading cause of cancer related deaths after metastasis. The authors should discuss the role of CAT, VTE and how chemotherapy increases this further in CML. The title of the review is also slightly confusing as the authors don't discuss mouse models to a huge degree and refer mainly to the ponatinib-induced thrombosis in mouse models. Is there any clinical information relating to the incidences of thrombosis in patients who receive ponatinib in comparison to other chemotherapies? The review mainly focuses on the cardiovascular effects of ponatinib and not its potential to exacerbate cancer associated thrombosis in patients with CML and ALL. More information is needed on this before publication. 

Author Response

Q: Cancers themselves are extremely pro-thrombotic, with cancer associated thrombosis the 2nd leading cause of cancer related deaths after metastasis.

A: Thank you for your comment.  We agree with this statement.  Our topic is on TKI-induced cardiovascular events and how these agents, in particular ponatinib, contribute to a patient disease substratum that by itself is prothrombotic.

Q: The authors should discuss the role of CAT, VTE and how chemotherapy increases this further in CML.

A: We appreciate the reviewer’s comment.  This entire review is a focused presentation on how TKIs used to treat CML and BCR-ABL positive ALL, particularly ponatinib, are themselves associated with increased cardiovascular events including thrombosis.  What is not known is how CML itself contributes to increased thrombosis.  It has been known for nearly 50 years that myeloid cells in CML have increased tissue factor.  However, that point has not been translated into experimental and clinical studies to show a relationship to thrombosis risk.  We intend to examine this issue with transgenic mice that have CML in the near future.  However, in the present review our focus is on these drugs and the clinical and experimental mechanisms by which they contribute to clinical thrombotic events.

Q: The title of the review is also slightly confusing as the authors don't discuss mouse models to a huge degree and refer mainly to the ponatinib-induced thrombosis in mouse models.

A: We agree with the reviewer’s comment.  We have revised the title in next version of the manuscript to “Ponatinib and Other Tyrosine Kinase Inhibitors in Thrombosis.”  Thank you for calling our attention to this.

Q: Is there any clinical information relating to the incidences of thrombosis in patients who receive ponatinib in comparison to other chemotherapies? The review mainly focuses on the cardiovascular effects of ponatinib and not its potential to exacerbate cancer associated thrombosis in patients with CML and ALL. More information is needed on this before publication.

A: Ponatinib use was the canary in the mine for recognizing that CML treatments with TKIs leads to increased cardiovascular events including arterial and venous thrombosis (Ref#88).  This information sparked interest in the field and additional studies have been performed.  In new Ref #98 by Jain P et al., Blood Advances 3: 851, 2019, 6 clinical trials containing 531 patients were reviewed for cardiovascular adverse events including stroke and myocardial infarction associated with front-line TKIs including ponatinib.  In this study, overall 237/531 patients (45%) developed CV adverse events (AEs).  Hypertension was most common (175/531) (33%).  Among the agents, ponatinib shows the highest incidence ratio (IR) (95% CI) for CV-AEs.  Concerning thrombosis, the incidence of thrombosis with the frontline TKIs are as follows: ponatinib 16.3%, dasatinib 16%, nilotinib 9.3%, and imatinib 4.7%.  This information has been added to the text at lines 266-278 in the revised manuscript.

Reviewer 3 Report

This review provides an up-to-date summary of the use of tyrosine kinase inhibitors, targeting Bcr-Abl1, in the treatment of CML and ALL patients. A particular focus of the review is the third generation tyrosine kinase inhibitor, ponatinib, and the mechanism of its off-target side-effect of increased risk of thrombosis. The review is well organised and the field is comprehensively described and well referenced. A major issue that needs address is that the manuscript would benefit from extensive proofreading. In places the English usage is awkward, some sentences have no verb, there is inconsistent use of the definitive and indefinitive article, errors in subject-verb agreement, as well as other grammatical and typographical errors.

1. In section 6, mechanism of ponatinib-induced thrombosis, it would be useful to provide some commentary on why there are apparently opposing  effects of ponatinib on platelet function in vitro versus in vivo.

2. Lines 336, 337, 344, 350, and 352: reference 97 should read reference 99. References should be checked throughout the manuscript.

3. In figure 2, there is no depiction of Par4 or its signalling pathway. The figure and figure title should be modified to remove reference to Par4.

Author Response

Q: A major issue that needs address is that the manuscript would benefit from extensive proofreading. In places the English usage is awkward, some sentences have no verb, there is inconsistent use of the definitive and indefinitive article, errors in subject-verb agreement, as well as other grammatical and typographical errors.

A: Thank you for your comments.  The article has been carefully proof read before resubmission to correct these mistakes.

Q: 1. In section 6, mechanism of ponatinib-induced thrombosis, it would be useful to provide some commentary on why there are apparently opposing effects of ponatinib on platelet function in vitro versus in vivo.

A: Thank you for this question. Actually, there may be no real inconsistency between the Loren CP et al. and Merkulova A et al. studies.  The differences may be due to the final concentration of ponatinib in the experiments.  In the Loren CP et al in vitro investigation, Ref 99, the final concentrations of ponatinib on the Lyn signaling experiments was between 0.1-1 mM.  It is possible that at these concentrations, both the regulatory p-LynY507 and constitutively activate p-LynY396 sites are inhibited and the GPVI activation pathway is blocked.  In our in vivo experiments, Ref 100, the maximal possible concentration of ponatinib achievable was less that 33 nM.  We have as yet not performed this experiment, but we postulate that the lower ponatinib concentration may have only blocked the regulatory p-LynY507 site and not the constitutively activated p-LynY396 site.  The basis for this assessment is that in studies on murine and human platelet samples from individuals on therapeutic doses of ponatinib that lead to plasma concentrations between 2-33 nM, we only see the higher regulatory site p-LynY507 blocked.  Since in platelets p-LynY396 is constitutively active, inhibition of only p-LynY507 results in p-Lyn changing from a closed to an open conformation to result in a “hyperactive” downstream GPVI activation pathway.  These phosphorylation studies are unpublished and will be in a future manuscript.  In the present manuscript we added that the concentration of ponatinib achieved in vivo is lower than that used in the Loren experiments.  This new information is found on lines 365-370 in the revised manuscript.

Q: Lines 336, 337, 344, 350, and 352: reference 97 should read reference 99. References should be checked throughout the manuscript.

A: Thank you very much for this.  We made these corrections

Q: In figure 2, there is no depiction of Par4 or its signaling pathway. The figure and figure title should be modified to remove reference to Par4.

A: Thank you for pointing this error out.  Figure 2 has been removed from the manuscript.

Round 2

Reviewer 1 Report

Well done

Reviewer 2 Report

I thank the authors for their responses. I would of liked more discussion on the role of TKI induced thrombosis from a clinical VTE perspective but accept the authors responses to my comments.